# DRUPI: Dataset Reduction Using Privileged Information

## Abstract

Dataset reduction (DR) seeks to select or distill samples from large datasets into smaller subsets while preserving performance on target tasks. Existing methods primarily focus on pruning or synthesizing data in the same format as the original dataset, typically the input data and corresponding labels. However, in DR settings, we find it is possible to synthesize more information beyond the data-label pair as an additional learning target to facilitate model training. In this paper, we introduce Dataset Reduction Using Privileged Information (DRUPI), which enriches DR by synthesizing privileged information alongside the reduced dataset. This privileged information can take the form of feature labels or attention labels, providing auxiliary supervision to improve model learning. Our findings reveal that effective feature labels must balance between being overly discriminative and excessively diverse, with a moderate level proving optimal for improving the reduced dataset's efficacy. Extensive experiments on ImageNet, CIFAR-10/100, and Tiny ImageNet demonstrate that DRUPI integrates seamlessly with existing dataset reduction methods, offering significant performance gains. *Code is included in the supplementary material and will be released.*

## 1 Introduction

Dataset Reduction (DR) has attracted considerable attention in recent years, with the primary aim of compressing large datasets into smaller subsets while maintaining comparable statistical performance. Existing methods for DR can be broadly classified into two main categories: *coreset selection* and *dataset distillation*. Coreset selection methods focus on selecting a subset of samples from the original dataset (Har-Peled & Mazumdar, 2004; Welling, 2009; Toneva et al., 2018), while dataset distillation involves synthesizing unseen samples from the dataset (Wang et al., 2018; Zhao et al., 2020; Zhao & Bilen, 2022; Cazenavette et al., 2022; Wang et al., 2024).

In typical real-world scenarios, training models for target tasks is generally constrained to input data (*e.g.*, images) and their corresponding labels, as these are the most readily available elements. While existing DR methods have shown strong performance (Wang et al., 2018; Zhao et al., 2020; Zhao & Bilen, 2022; Cazenavette et al., 2022; Yin et al., 2023), they typically do so by compressing datasets in the same or similar format, such as the conventional data-label structure. Even advanced dataset distillation techniques, which re-parameterize images or labels to create alternative representations (Kim et al., 2022; Zhao et al., 2023; Deng & Russakovsky, 2022; Liu et al., 2022; Wei et al., 2024), are limited by this conventional framework. As illustrated in Figure 1(a), this reliance on fixed data-label structures restricts the capacity of such methods to incorporate richer information that could further enhance model training and improve generalization.

In fact, DR settings offer the potential to create more diverse compressed datasets that extend beyond simple input data $x_i$ and labels $y_i$, incorporating richer forms of information. A notable example is the concept of *privileged information*, first introduced in the context of statistical learning (Vapnik & Vashist, 2009; Pechyony & Vapnik, 2010). Figure 2 provides a illustration for privileged information. Let us consider a more concrete example, where $x_i$ might represent a biopsy image, and the privileged information $f_i^\star$ for $x_i$ could be the oncologist's written assessment of the image (Vapnik et al., 2015). The label $y_i$ would then indicate whether the tissue in the image is malignant or benign. By leveraging this privileged information $f_i^\star$, a medical expert can make more informed decisions, benefiting from additional insights that improve diagnostic accuracy.

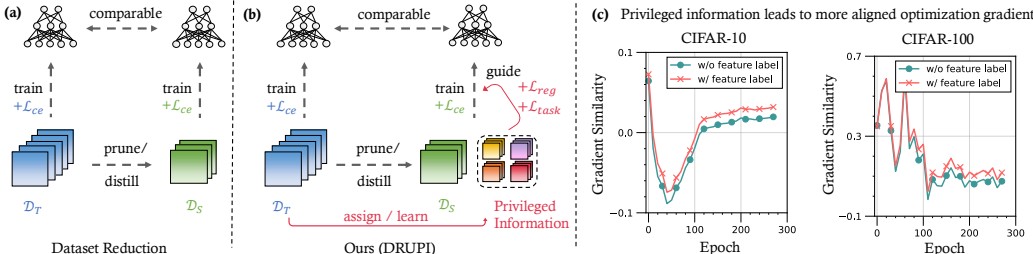

Figure 1: A comparison between conventional dataset reduction pipelines and our proposed DRUPI framework. (a) Previous dataset reduction methods distill or select a subset $\mathcal{D}_\mathcal{S}$ from the original dataset $\mathcal{D}_\mathcal{T}$, maintaining the original "data-label" structure. (b) In contrast, DRUPI synthesizes auxiliary privileged information from $\mathcal{D}_\mathcal{T}$, enriching further supervision to models trained on the reduced subset $\mathcal{D}_\mathcal{S}$. (c) Cosine similarity between the gradients of a pre-trained model on synthetic datasets w/ and w/o privileged information (feature labels) and the real dataset. Synthetic datasets are generated using DC with 10 IPC. We used the same pre-trained ConvNet for gradient extraction.

However, none of the existing methods compress the original dataset beyond the traditional data-label structure or synthesize privileged information for auxiliary supervision. To address this gap, we introduce a novel approach that, for the first time, **incorporates not only images and labels but also privileged information**. Our method, called **D**ataset **R**eduction **U**sing **P**rivileged **I**nformation (DRUPI), is illustrated in Figure 1(b). We primarily synthesize feature labels for the reduced dataset, as these labels capture richer, high-dimensional information, enhancing dataset quality. These feature labels generalize effectively across various neural network architectures and provide a unified representation of latent statistics across multiple models, offering additional supervision during training. Additionally, we propose a more efficient form of feature labels, *i.e.*, attention labels. As shown in Figure 1(c), the incorporation of privileged information produces gradients more aligned with those of the original dataset, ultimately improving the model's generalization capabilities.

A key challenge for DRUPI lies in determining appropriate feature labels to synthesize for the reduced dataset. To address this, we employ an arbitrary methods of dataset distillation to synthesize the feature labels. During each step of the bi-level optimization, we match the statistical information of models trained on reduced datasets with and without feature labels. Furthermore, we observed that synthesized feature labels cannot be overly discriminative or diverse, which degrade the overall quality of the reduced dataset. This finding suggests that an optimal balance between discriminability and diversity is crucial for synthesizing effective feature labels. Our contributions are summarized as follows:

1. We propose a new paradigm, *i.e.*, DRUPI, for dataset reduction. In particular, privileged information, such as feature labels, can be synthesized in addition to traditional data-label pairs. This privileged information provides additional generative supervision during model training, thereby improving the generalization ability of the reduced dataset.

2. We observe that effective feature labels should balance the trade-off between diversity and discriminability. Overly discriminative feature labels, such as those directly extracted from a pre-trained neural network, can even degrade the quality of the reduced dataset.

3. We further provide a theoretical analysis of our DRUPI pipeline based on VC theory (Vapnik, 1998) from statistical learning, which rigorously ensures its effectiveness.

4. Our experiments demonstrate that DRUPI can be seamlessly integrated into state-of-the-art DR methods. Particularly, for coreset selection methods, applying DRUPI to Herding on CIFAR10 (with a fraction of 0.4%) improves performance by 24.3%, while applying it to K-center in cross-architecture evaluations leads to an improvement of up to 23.4%. For dataset distillation methods, integrating DRUPI with the DC method on CIFAR100 (with 10 images per class) yields a 4% improvement, and further cross-architecture evaluations of DC show gains of up to 18.3%.

## 2 BACKGROUND AND RELATED WORK

Similar to prior works on dataset reduction (DR), we focus on classification tasks (Welling, 2009; Zhao et al., 2020; Shin et al., 2023). Consider a multi-classification problem. Let $\mathcal{X} \in \mathbb{R}^d$ represent

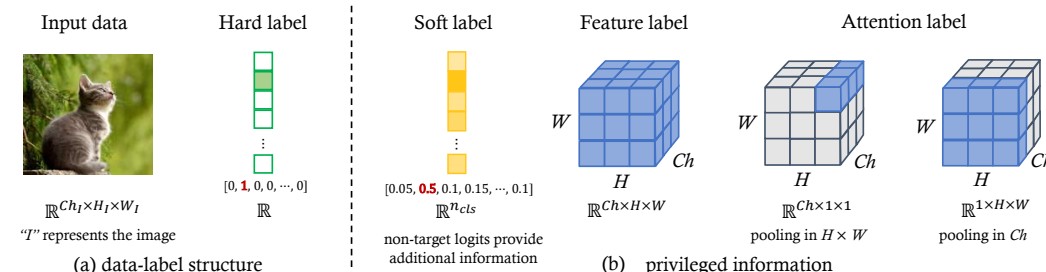

Figure 2: Comparison between (a) the traditional "data-label" structure and (b) Different forms of privileged information. Non-target classes of soft labels provide additional information, can be considered a form of privileged information. Feature labels encapsulate high-dimensional information. Attention labels are obtained by applying average pooling to feature labels.

the input space and $\mathcal{Y}$ denote the set of possible labels. Our dataset, $\mathcal{D}_\mathcal{T} = \{(x_i, y_i)\}_{i=1}^n \subseteq \mathcal{X} \times \mathcal{Y}$, consists of $n$ training samples, where each $x_i \in \mathcal{X}$ is an input vector and $y_i \in \mathcal{Y}$ is its corresponding label. In typical DR settings, the goal is to obtain a smaller dataset, $\mathcal{D}_\mathcal{S}$ with size $m$, where $m \ll n$. DR methods are typically divided into two categories: *coreset selection*, where the reduced dataset is a subset of the original, and *dataset distillation*, where the reduced dataset consists of synthesized data not present in the original set but learned through optimization.

**Coreset selection techniques.** Coreset selection techniques are designed to identify a representative subset $\mathcal{D}_\mathcal{S}$ from the complete dataset $\mathcal{D}_\mathcal{T}$. This process typically seeks to optimize a criterion that quantifies the informativeness of $\mathcal{D}_\mathcal{S}$, which matches that of the original dataset $\mathcal{D}_\mathcal{T}$. The informativeness can be gauged by various metrics, including gradients (Paul et al., 2021; Mirzasoleiman et al., 2020; Killamsetty et al., 2021a), loss values (Toneva et al., 2018), predictive uncertainties (Coleman et al., 2019b), proximity to decision boundaries (Ducoffe & Precioso, 2018; Margatina et al., 2021), and the sharpness of the learned model (Shin et al., 2023).

**Dataset distillation methods.** Dataset distillation approaches offer an alternative strategy by focusing on synthesizing a distilled dataset $\mathcal{D}_\mathcal{S}$ from the original dataset $\mathcal{D}_\mathcal{T}$, rather than directly selecting a subset. This is typically accomplished through a bi-level optimization process that aligns the performance of $\mathcal{D}_\mathcal{S}$ with that of $\mathcal{D}_\mathcal{T}$. A distance metric $\mathbf{D}$ is employed to quantify the statistical divergence between datasets, guiding the learning of $\mathcal{D}_\mathcal{S}$ via gradient descent. Specifically, the distilled dataset is updated as follows: $\mathcal{D}_\mathcal{S} \leftarrow \mathcal{D}_\mathcal{S} - \eta \cdot \nabla_{\mathcal{D}_\mathcal{S}} \mathbf{D}(\mathcal{D}_\mathcal{S}, \mathcal{D}_\mathcal{T})$. The choice of distance metric $\mathbf{D}$ is versatile and can encompass various aspects such as gradients (Zhao et al., 2020; Lee et al., 2022; Zhao & Bilen, 2021), feature representations (Zhao & Bilen, 2022; Sajedi et al., 2023), training trajectories (Cazenavette et al., 2022; Du et al., 2023; Cui et al., 2023; Guo et al., 2023), and kernel information (Nguyen et al., 2020; 2021; Zhou et al., 2022).

In addition to direct performance matching, certain methodologies endeavor to re-parameterize input data to enhance compression efficiency. Techniques employed in this context include exploiting data regularity, as discussed in various studies (Kim et al., 2022; Zhao et al., 2023; Son et al., 2024), factorizing images to capture intrinsic structures (Liu et al., 2022; Deng & Russakovsky, 2022), and employing sparse coding to represent data effectively (Wei et al., 2024).

## 3 METHOD

### 3.1 DETERMINING PRIVILEGED INFORMATION

Although prior research on dataset reduction has shown impressive results, it mainly focuses on generating reduced datasets in conventional data-label formats, as depicted in Figure 2(a). However, more informative data, such as privileged information, can be utilized to enhance both the utility and representational performance of reduced datasets, as illustrated in Figure 2(b). Below, we briefly explore several forms of privileged information that can be incorporated to achieve this goal.

**Soft labels.** We first claim that soft labels are a form of privileged information, as they offer richer insight into how an expert model interprets predictions, providing soft probabilities rather than a single hard label. Specifically, the non-target logits can be viewed as additional information for

supervision. Several works have previously explored the effectiveness of soft labels in dataset reduction (Guo et al., 2023; Cui et al., 2023; Bohdal et al., 2020; Qin et al., 2024). However, while soft labels enhance the available information, they are limited to low-dimensional discriminative representations and fail to capture more complex, high-dimensional statistics. Moreover, they do not fundamentally alter the data-label structure of the reduced dataset representation.

**Feature labels.** Beyond soft labels, we propose feature labels as a more effective form of privileged information. These labels, derived from unified intermediate representations across well-trained models, encapsulate rich, high-dimensional latent statistics. By providing additional supervision, feature labels enhance model training on datasets with privileged information. Unlike approaches that focus primarily on soft labels, assigning a unified feature label to each input enriches the supervision signal for downstream tasks, effectively addressing the limitations of prior methods.

**Attention labels.** Alongside feature labels, we propose attention labels as an alternative form of privileged information that provides a more memory-efficient representation. Attention labels can be derived from either spatial or channel attention of feature labels (Woo et al., 2018). For example, given a feature label of size $Ch \times H \times W$, spatial attention reduces the $Ch$ dimensions through pooling operations (*e.g.*, average pooling, max pooling), resulting in an attention label of size $1 \times H \times W$. Similarly, channel attention applies pooling along the $H \times W$ dimension to produce a reduced representation, *i.e.*, $Ch \times 1 \times 1$. Both feature and attention labels are valuable, with attention labels offering a more efficient representation but with a possible trade-off in the richness of information. We provide more discussion on attention labels in Appendix B.2.

In this work, we primarily focus on generating additional *feature labels*[1] for the reduced dataset, as these forms of privileged information provide more complementary and useful insights for model training. However, privileged information can take various forms beyond attention and feature labels. Depending on the task and the model architecture, other types of information, such as learned embeddings, domain-specific signals, or task-related metadata, could be equally beneficial in enhancing the informativeness and performance of reduced datasets. The flexibility to incorporate different kinds of privileged information allows us to tailor the dataset to specific needs and maximize the potential of the reduced data.

## 3.2 SYNTHESIZING PRIVILEGED INFORMATION

In this section, we discuss the process of generating privileged information for a given reduced dataset $\mathcal{D}_\mathcal{S} = \{(\tilde{x}_i, \tilde{y}_i)\}_{i=1}^m$, with the goal of obtaining a more informative dataset $\mathcal{D}_\mathcal{S}^\star = \{(\tilde{x}_i, \tilde{y}_i, f_i^\star)\}_{i=1}^m$. Here, $\mathcal{D}_\mathcal{S}$ is the reduced dataset of a larger dataset $\mathcal{D}_\mathcal{T} = \{(x_i, y_i)\}_{i=1}^n$, where $m \ll n$. Our primary focus is on incorporating feature labels as the form of privileged information. While various methods can be employed to synthesize privileged information, they can generally be categorized into two strategies: *direct assignment* and *learning-based methods*.

**Direct Assignment of Feature Labels.** A straightforward approach to obtaining feature labels is by using a pre-trained model, *e.g.*, $g_\mathcal{T}$, to extract intermediate features for each input data $\tilde{x}_i \in \mathcal{D}_\mathcal{S}$. Specifically, this is formalized as $f_i^\star = g_\mathcal{T}(\tilde{x}_i)$, resulting in an extended dataset represented as $\mathcal{D}_\mathcal{S}^\star = (\tilde{x}_i, \tilde{y}_i, f_i^\star)$. This method is computationally efficient but relies heavily on the generalization ability of the pre-trained model $g_\mathcal{T}$. While the feature labels, which capture the implicit biases of $g_\mathcal{T}$, may enhance the quality of the reduced dataset, they could also introduce potential drawbacks. In fact, directly assigned feature labels are often overly discriminative, reducing diversity. However, we find that suitable feature labels should strike a balance between these two properties.

**Learning Feature Labels.** A more robust approach to obtaining feature labels is through learning-based methods. Many dataset distillation techniques can be adapted for learning feature labels. For instance, we can employ the Dataset Condensation (DC) (Zhao et al., 2020) method as an illustrative example to guide the process of learning synthetic feature labels. Suppose we have a learned synthetic dataset $\mathcal{D}_\mathcal{S}$. In the typical DC method, we initialize a random model $g$ with parameters $\theta = \theta_0$ and train it for $T$ epochs on both $\mathcal{D}_\mathcal{T}$ and $\mathcal{D}_\mathcal{S}$ separately. The synthetic dataset $\mathcal{D}_\mathcal{S}$ is updated by matching the category gradients between the two datasets, which can be expressed as follows[2]:

---

[1] We consider attention labels as a specific form of feature labels. Therefore, for simplicity, we use the term "feature labels" as a unified description for both feature labels and attention labels.

[2] We ignore the category symbol for simplicity.

$$\mathcal{D}_{\mathcal{S}} = \arg\min_{\mathcal{D}_{\mathcal{S}}} \mathbb{E}_{\theta_0 \sim P_\theta} \left[ \sum_{t=0}^{T} \mathbf{D} \left( \nabla_\theta L(\mathcal{D}_{\mathcal{S}}; \theta_t), \nabla_\theta L(\mathcal{D}_{\mathcal{T}}; \theta_t) \right) \right]$$

$$\text{where} \quad L(\mathcal{D}_{\mathcal{T}}; \theta_t) = \mathbb{E}_{(x_i, y_i) \in \mathcal{D}_{\mathcal{T}}} \ell_{ce} \left[ (y_i, \sigma(g(x_i; \theta_t))) \right], \tag{1}$$

$$\text{and} \quad L(\mathcal{D}_{\mathcal{S}}; \theta_t) \triangleq \mathcal{L}_{cls} = \mathbb{E}_{(\tilde{x}_i, \tilde{y}_i) \in \mathcal{D}_{\mathcal{S}}} \left[ \ell_{ce} \left( \tilde{y}_i, \sigma(g(\tilde{x}_i; \theta_t)) \right) \right],$$

where $\ell_{ce}(\cdot, \cdot)$ denotes the cross-entropy (CE) loss, and $\sigma(\cdot)$ represents the softmax function. In contrast, we aim to match the performance between $\mathcal{D}_{\mathcal{T}}$ and $\mathcal{D}_{\mathcal{S}}^\star$, where privileged information is synthesized to capture additional informativeness. Let $\ell_{mse}$ represent the mean square error (MSE) loss, and let $\psi(\cdot)$ denote the intermediate output of model $g$, *i.e.*, $g = \psi \circ \kappa$, where $\kappa(\cdot)$ is the classifier component of $g$. Therefore, our objective becomes:

$$\mathcal{D}_{\mathcal{S}}^\star = \arg\min_{\mathcal{D}_{\mathcal{S}}^\star} \mathbb{E}_{\theta_0 \sim P_\theta} \left[ \sum_{t=0}^{T} \mathbf{D}(\nabla_\theta L_c(\mathcal{D}_{\mathcal{S}}^\star; \theta_t), \nabla_\theta L_c(\mathcal{D}_{\mathcal{T}}; \theta_t)) \right], \tag{2}$$

$$\text{where} \quad L(\mathcal{D}_{\mathcal{S}}^\star; \theta_t) \triangleq \mathcal{L}_{cls} + \lambda_{reg} \cdot \mathcal{L}_{reg},$$

$$\mathcal{L}_{cls} = \mathbb{E}_{(\tilde{x}_i, \tilde{y}_i) \in \mathcal{D}_{\mathcal{S}}^\star} \left[ \ell_{ce} \left( y_i, \sigma(g(\tilde{x}_i; \theta_t)) \right) \right],$$

$$\text{and} \quad \mathcal{L}_{reg} = \mathbb{E}_{(\tilde{x}_i, \tilde{y}_i, f_i^\star) \in \mathcal{D}_{\mathcal{S}}^\star} \left[ \ell_{mse} \left( f_i^\star, \psi(\tilde{x}_i; \theta_t) \right) \right], \tag{3}$$

where $\lambda_{reg}$ is a hyper-parameter to determine the scale of using privileged information. In addition to DC, other dataset distillation methods can also be employed to synthesize feature labels $f_i^\star$. We provide further resuls on coreset selection methods like Herding (Welling, 2009), K-center (Har-Peled & Mazumdar, 2004), Forgetting (Toneva et al., 2018), and dataset distillation methods like DC (Zhao et al., 2020), MTT (Cazenavette et al., 2022), and DATM (Guo et al., 2023).

In addtional, we introduce additional supervision to enhance the discriminative power of these feature labels while preserving their diversity.

• *Task-oriented synthesization.* To improve the discriminative power of the feature labels, we adopt a task-oriented approach by feeding the synthesized feature labels into the classifier of the model used to extract gradients during bi-level optimization. We achieve this by performing additional CE loss between the feature label $f_i^\star$'s prediction and the ground-truth label $\tilde{y}_i$. Specifically, we have

$$\mathcal{L}_{task} = \mathbb{E}_{(\tilde{f}_i^\star, \tilde{y}_i) \in \mathcal{D}_{\mathcal{S}}^\star} \left[ \ell_{ce} \left( \tilde{y}_i, \sigma(\kappa(f_i^\star; \theta_t)) \right) \right]. \tag{4}$$

This allows the feature labels to contribute directly to the final prediction, ensuring they become better aligned with the task at hand. The scale of task supervision is controlled by the hyper-parameter $\lambda_{task}$. Our observations indicate that the **preferred feature labels should strike a balance between discriminability and diversity, *i.e.*, they should neither be overly discriminative nor completely lack discriminative power**. As shown in Figure 3(a)(c), increasing $\lambda_{task}$ tends to cluster the feature labels, reducing their diversity while increasing their discriminability. We find that the optimal feature labels are not achieved with the highest task supervision. Instead, a moderate level of task supervision, as shown in Figure 3(b), strikes the right balance between diversity and discriminability. This also suggests a possible explanation for the drawbacks of directly assigning feature labels from a well-trained model, as these labels tend to be overly discriminative.

• *Versatility synthesization.* Beyond task-specific supervision, we aim to preserve the generative versatility of the feature labels, *i.e.*, a set of feature labels $f_i^\star \in F_i$, where $F_i$ represents the possible feature label set for input $x_i$. This approach involves synthesizing multiple feature labels for a single data-label pair. Appropriate versatility enhancement ensures that the synthesized feature labels remain informative across different tasks and applications, providing a richer and more comprehensive representation of the data. When using multiple feature labels, we primarily employ two strategies: randomly selecting one or using the average feature label from $F_i$. Further discussions demonstrating the benefits of this versatility are presented in Figure 4(a) and Appendix B.3.

We define the overall loss function for a model trained on the reduced dataset $\mathcal{D}_{\mathcal{S}}^\star$ as follows:

$$L(\mathcal{D}_{\mathcal{S}}^\star; \theta_t) = \mathcal{L}_{cls} + \mathbb{E}_{f_i^\star \in F_i} \left[ \lambda_{reg} \cdot \mathcal{L}_{reg} + \lambda_{task} \cdot \mathcal{L}_{task} \right], \tag{5}$$

where $F_i$ denotes the feature label set, containing multiple $f_i^\star$ for a single data-label pair $(\tilde{x}_i, \tilde{y}_i)$. The pseudocode for learning privileged information is provided in Appendix D.

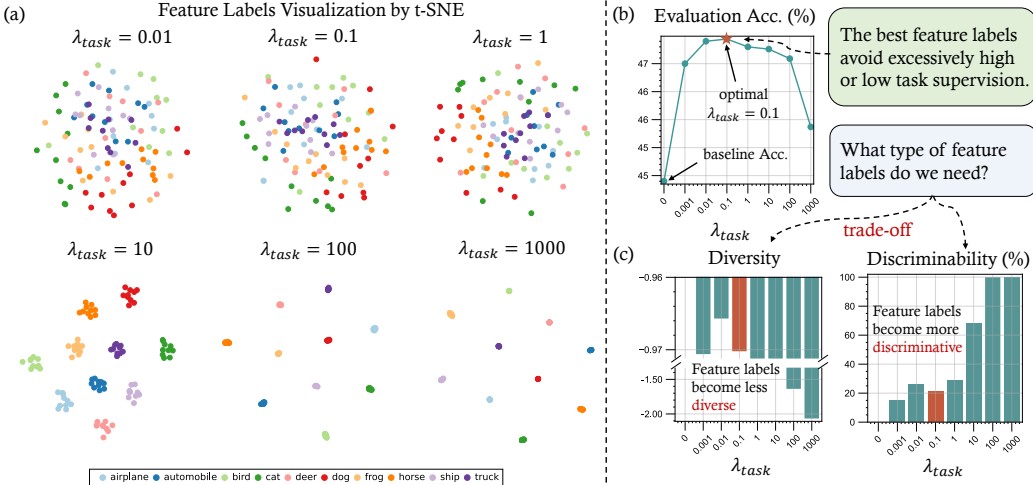

Figure 3: Feature labels learned under varying levels of task supervision. (a) t-SNE visualization of feature labels learned with different task supervision coefficients $\lambda_{task}$. (b) The most effective feature labels are produced with a moderate level of task supervision, avoiding excessively high or low supervision. (c) Increasing task supervision makes the feature labels more discriminative but less diverse. Diversity is measured by the negative mutual information between the feature labels and the ground truth labels, while discriminability is measured by the classification accuracy of a linear classifier trained on the feature labels.

## 3.3 LEARNING USING PRIVILEGED INFORMATION

We have previously discussed how to synthesize appropriate privileged information $f_i^\star$ for a given reduced dataset $\mathcal{D}_\mathcal{S}$ and extend it into $\mathcal{D}_\mathcal{S}^\star$. We now focus on leveraging the new reduced dataset $\mathcal{D}_\mathcal{S}^\star$ to enhance a model's performance on unseen test data. Following the *learning using privileged information* (LUPI) framework (Pechyony & Vapnik, 2010; Lopez-Paz et al., 2015), we incorporate the additional privileged information $f_i^\star$ during training to build a classifier that outperforms those trained solely on the regular reduced dataset $\mathcal{D}_\mathcal{S}$.

Given an arbitrary model $h$ with parameters $\theta \in \Theta$, we now formally discuss how to train $h$ with synthesized feature labels. The same loss function $L(\mathcal{D}_\mathcal{S}^\star)$, as shown in Eq. (5), is applied to train $h$, with hyper-parameters (*e.g.*, $\lambda_{reg}, \lambda_{task}$) kept consistent for fairness.

$$\theta^\star = \underset{\theta \in \Theta}{\arg\min}\, L(\mathcal{D}_\mathcal{S}^\star; \theta) \qquad (6)$$

Besides feature labels, we can store attention labels, which can be generated by performing average pooling on the spatial or channel dimensions of learned feature labels. Attention labels contain more condensed information, which can further reduce storage cost. During LUPI, we apply the same pooling strategy for intermediate features of the given model to calculate the MSE loss between the intermediate features and given feature labels.

## 3.4 THEORETICAL ANALYSIS

We propose a theoretical analysis to elucidate the mechanisms by which the DRUPI framework enhances the quality of reduced datasets. Let $g_\mathcal{T} \in \mathcal{G}_\mathcal{T}$ denote the oracle function for the original dataset $\mathcal{D}_\mathcal{T}$, and $|\cdot|_C$ represent a function class capacity measure, *i.e.*, a measure of model performance. Consider two models: one trained on the pure reduced dataset $\mathcal{D}_\mathcal{S}$, represented by $g_\mathcal{S} \in \mathcal{G}_\mathcal{S}$, and another trained on the reduced dataset with privileged information, $\mathcal{D}_\mathcal{S}^\star$, denoted by $g_{\mathcal{S}^\star} \in \mathcal{G}_{\mathcal{S}^\star}$.

We begin by briefly reviewing the well-known VC theory (Vapnik, 1998), a fundamental analytical tool in statistical learning theory that forms the basis of our theoretical framework. The VC-dimension defines the performance of a model to limited data points. Specifically, for a model $g$ belonging to a function class $\mathcal{G}$, with a finite VC-dimension $|\mathcal{G}|_{\text{VC}}$, the expected error $R(g)$ with

probability $1 - \delta$ is bounded as follows:

$$R(g) \leq R_m(g) + O\left(\left(\frac{|\mathcal{G}|_{\mathrm{VC}} - \log \delta}{m}\right)^{\alpha}\right), \tag{7}$$

where the $O(\cdot)$ term is the estimation error, and $R_m(g)$ is the training error over $m$ data points, and $\alpha$ lies between $\frac{1}{2}$ and 1. The parameter $\alpha$ represents the difficulty of the task. For more difficult, non-separable tasks, $\alpha \approx \frac{1}{2}$, yielding a slower learning rate of $O(m^{-1/2})$. For easier, separable tasks, where the model makes no training errors, $\alpha \approx 1$, yielding a faster learning rate of $O(m^{-1})$. Given a student learning from a fixed dataset of size $m$, a good teacher model can ease the learning process by accelerating the learning rate from $O(m^{-1/2})$ to $O(m^{-1})$.

Next, we provide theoretical analysis for DRUPI, showing how incorporating privileged information accelerates the learning process and improves the quality of the reduced dataset. Building on the top of (Lopez-Paz et al., 2015), we extend the results in dataset reduction scenarios. First, assume that the model trained on the pure reduced dataset $g_{\mathcal{S}}$ learns the true function $g_{\mathcal{T}}$ at a slower rate $\alpha_{\mathcal{S}}$:

$$R(g_{\mathcal{S}}) - R(g_{\mathcal{T}}) \leq O\left(\frac{|\mathcal{G}_{\mathcal{S}}|_{\mathrm{C}}}{m^{\alpha_{\mathcal{S}}}}\right) + \varepsilon_{\mathcal{S}}, \tag{8}$$

where $\varepsilon_{\mathcal{S}}$ is the approximation error of $\mathcal{G}_{\mathcal{S}}$ with respect to $g_{\mathcal{T}} \in \mathcal{G}_{\mathcal{T}}$. Second, assume that the model trained on the dataset with privileged information $g_{\mathcal{S}^{\star}}$ learns at a faster rate $\alpha_{\mathcal{S}^{\star}}$:

$$R(g_{\mathcal{S}^{\star}}) - R(g_{\mathcal{T}}) \leq O\left(\frac{|\mathcal{G}_{\mathcal{S}^{\star}}|_{\mathrm{C}}}{m^{\alpha_{\mathcal{S}^{\star}}}}\right) + \varepsilon_{\mathcal{S}^{\star}}, \tag{9}$$

where $\varepsilon_{\mathcal{S}^{\star}}$ is the approximation error of $\mathcal{G}_{\mathcal{S}^{\star}}$ with respect to $g_{\mathcal{T}}$. Finally, assume that the performance difference $g_{\mathcal{S}}$ learns from the model with privileged information $g_{\mathcal{S}^{\star}}$ is

$$R(g_{\mathcal{S}}) - R(g_{\mathcal{S}^{\star}}) \leq O\left(\frac{|\mathcal{G}_{\mathcal{S}}|_{\mathrm{C}}}{m^{\alpha}}\right) + \varepsilon, \tag{10}$$

where $\varepsilon$ is the approximation error of $\mathcal{G}_{\mathcal{S}}$ with respect to $g_{\mathcal{S}^{\star}}$, and $\frac{1}{2} \leq \alpha \leq 1$. Combining Eq. (9) and Eq. (10), the learning rate for the model without privileged information learning the oracle function $g_{\mathcal{T}}$ is then given by

$$R(g_{\mathcal{S}}) - R(g_{\mathcal{T}}) = R(g_{\mathcal{S}}) - R(g_{\mathcal{S}^{\star}}) + R(g_{\mathcal{S}^{\star}}) - R(g_{\mathcal{T}})$$
$$\leq O\left(\frac{|\mathcal{G}_{\mathcal{S}}|_{\mathrm{C}}}{m^{\alpha}}\right) + \varepsilon + O\left(\frac{|\mathcal{G}_{\mathcal{S}^{\star}}|_{\mathrm{C}}}{m^{\alpha_{\mathcal{S}^{\star}}}}\right) + \varepsilon_{\mathcal{S}^{\star}} \leq O\left(\frac{|\mathcal{G}_{\mathcal{S}}|_{\mathrm{C}}}{m^{\alpha}} + \frac{|\mathcal{G}_{\mathcal{S}^{\star}}|_{\mathrm{C}}}{m^{\alpha_{\mathcal{S}^{\star}}}}\right) + \varepsilon + \varepsilon_{\mathcal{S}^{\star}}, \tag{11}$$

where the final inequality arises because $\alpha \leq 1$. Therefore, for $\frac{1}{2} < \alpha \leq 1$ in dataset reduction settings, the inequality becomes:

$$O\left(\frac{|\mathcal{G}_{\mathcal{S}}|_{\mathrm{C}}}{m^{\alpha}} + \frac{|\mathcal{G}_{\mathcal{S}^{\star}}|_{\mathrm{C}}}{m^{\alpha_{\mathcal{S}^{\star}}}}\right) + \varepsilon + \varepsilon_{\mathcal{S}^{\star}} \leq O\left(\frac{|\mathcal{G}_{\mathcal{S}}|_{\mathrm{C}}}{m^{\alpha_{\mathcal{S}}}}\right) + \varepsilon_{\mathcal{S}}. \tag{12}$$

This inequality highlights the advantages of models trained with privileged information: training using privileged information exhibit lower generalization and approximation errors compared to those trained without privileged information. More importantly, it emphasizes that the privileged information is most beneficial in low-data regimes, which is the typical DR scenario. These benefits align with the principles of LUPI as outlined in (Vapnik et al., 2015; Lopez-Paz et al., 2015).

## 4 EXPERIMENTS

### 4.1 EXPERIMENTAL SETUP

In this section, we investigate the effectiveness of our proposed method, DRUPI, through a series of experiments on diverse datasets and tasks. We begin by evaluating the efficacy of DRUPI when applied to coreset selection and dataset distillation tasks. Specifically, we followed prior works to conduct experiments on CIFAR-10/100 (Krizhevsky et al., 2009) for coreset selection methods, where ResNet-18 (He et al., 2016) is utilized for extracting importance score. For the dataset distillation methods, we conducted experiments on CIFAR-10/100, Tiny ImageNet (Le & Yang, 2015), and subsets of ImageNet (Russakovsky et al., 2015).

Table 1: Application of DRUPI to representative pruning methods on CIFAR-10/100. We initialized the dataset with the baseline methods, and utilized DC for synthesizing feature labels.

| Dataset | CIFAR-10 | | | | | CIFAR-100 | | | | |
|---|---|---|---|---|---|---|---|---|---|---|
| Fraction (%) | 0.02 | 0.1 | 0.2 | 0.4 | 1 | 0.2 | 1 | 2 | 4 | 10 |
| Random | 13.5±0.4 | 20.0±0.5 | 27.1±0.6 | 35.8±0.6 | 43.0±0.5 | 4.3±0.2 | 9.5±0.2 | 14.5±0.2 | 19.5±0.4 | 29.5±0.3 |
| L-Conf | 10.7±0.4 | 10.5±0.4 | 10.8±0.4 | 17.9±0.4 | 23.1±0.6 | 2.1±0.1 | 3.6±0.1 | 6.6±0.2 | 9.0±0.2 | 16.4±0.3 |
| Entropy | 12.2±0.6 | 14.1±0.5 | 14.8±0.5 | 19.6±0.4 | 23.8±0.7 | 1.7±0.1 | 3.7±0.2 | 6.7±0.2 | 9.0±0.3 | 17.1±0.3 |
| Margin | 8.9±0.4 | 15.8±0.7 | 20.3±0.4 | 24.8±0.5 | 31.3±0.5 | 3.0±0.2 | 6.2±0.2 | 9.0±0.2 | 12.7±0.3 | 20.7±0.3 |
| Glister | 11.5±0.3 | 16.9±0.5 | 23.0±0.4 | 28.4±0.5 | 30.0±0.5 | 2.9±0.5 | 7.1±0.4 | 10.4±0.6 | 13.3±0.7 | 27.2±0.7 |
| Graig | 18.1±0.3 | 19.5±0.4 | 19.0±0.5 | 27.8±0.3 | 30.2±0.4 | 4.3±0.5 | 9.0±0.4 | 13.6±0.7 | 14.6±0.5 | 20.1±0.6 |
| Herding | 15.3±0.5 | 23.5±0.3 | 25.1±0.5 | 26.7±0.4 | 34.9±0.6 | 4.0±0.1 | 6.2±0.2 | 8.1±0.4 | 13.1±0.6 | 18.4±0.6 |
| +DRUPI | **27.9±0.6** | **37.3±0.6** | **45.8±0.7** | **51.0±0.4** | **54.0±0.6** | **14.0±0.3** | **20.4±0.5** | **25.5±0.4** | **28.9±0.6** | **31.4±0.6** |
| ↑ | 12.6 | 13.8 | 20.7 | 24.3 | 19.1 | 10.0 | 14.2 | 17.4 | 15.8 | 13.0 |
| k-Center | 16.4±0.6 | 22.4±0.6 | 23.1±0.5 | 30.4±0.4 | 36.7±0.5 | 4.8±0.2 | 6.7±0.4 | 10.0±0.5 | 15.9±1.1 | 21.8±1.0 |
| +DRUPI | **29.7±0.6** | **40.0±0.6** | **46.2±0.6** | **50.8±0.6** | **54.3±0.6** | **13.5±0.3** | **20.0±0.5** | **25.9±0.3** | **29.1±0.5** | **32.0±0.5** |
| ↑ | 13.3 | 17.6 | 23.1 | 20.4 | 17.6 | 8.7 | 13.3 | 15.9 | 13.2 | 10.2 |
| Forgetting | 15.3±0.6 | 19.1±0.7 | 23.9±0.7 | 26.9±0.7 | 39.5±0.5 | 4.1±0.1 | 7.8±0.3 | 10.4±0.5 | 14.1±0.6 | 22.3±0.4 |
| +DRUPI | **30.0±0.6** | **39.7±0.7** | **46.6±0.6** | **51.3±0.5** | **54.5±0.5** | **14.0±0.4** | **20.1±0.6** | **25.8±0.3** | **29.3±0.4** | **32.2±0.5** |
| ↑ | 14.7 | 20.6 | 22.7 | 24.4 | 15.0 | 9.9 | 12.3 | 15.4 | 15.2 | 9.9 |
| Full Dataset | 84.8±0.1 | | | | | 56.2±0.3 | | | | |

Table 2: Results of DRUPI on distillation methods across CIFAR-10/100, and Tiny ImageNet. We initialized reduced datasets with corresponding baseline methods, and synthesized feature labels for these datasets with the same baseline methods.

| Dataset | CIFAR-10 | | | CIFAR-100 | | | Tiny ImageNet | |
|---|---|---|---|---|---|---|---|---|
| IPC | 1 | 10 | 50 | 1 | 10 | 50 | 1 | 10 |
| Random | 15.4±0.3 | 31.0±0.5 | 50.6±0.3 | 4.2±0.3 | 14.6±0.5 | 33.4±0.4 | 1.4±0.1 | 5.0±0.2 |
| KIP | 49.9±0.2 | 62.7±0.3 | 68.6±0.3 | 15.7±0.2 | 28.3±0.1 | - | - | - |
| DM | 26.0±0.8 | 48.9±0.6 | 63.0±0.4 | 11.4±0.3 | 29.7±0.3 | 43.6±0.4 | 3.9±0.2 | 12.9±0.4 |
| DSA | 28.8±0.7 | 52.1±0.5 | 60.6±0.5 | 13.9±0.3 | 32.3±0.3 | 42.8±0.4 | - | - |
| DCC | 32.9±0.8 | 49.4±0.5 | 61.6±0.4 | 13.3±0.3 | 30.6±0.4 | 40.0±0.3 | - | - |
| DSAC | 34.0±0.7 | 54.5±0.5 | 64.2±0.4 | 14.6±0.3 | 33.5±0.3 | 39.3±0.4 | - | - |
| CAFE | 30.3±1.1 | 46.3±0.6 | 55.5±0.6 | 12.9±0.3 | 27.8±0.3 | 37.9±0.3 | - | - |
| IDM | 45.6±0.7 | 58.6±0.1 | 67.5±0.1 | 20.1±0.3 | 45.1±0.1 | **50.0±0.2** | - | - |
| DC | 28.3±0.5 | 44.9±0.5 | 53.9±0.5 | 12.8±0.3 | 25.2±0.3 | 29.8±0.3 | - | - |
| +DRUPI | 31.5±0.9 | 47.4±0.9 | 55.0±0.5 | 14.9±0.4 | 29.2±0.5 | 30.9±0.5 | - | - |
| ↑ | 3.2 | 2.5 | 1.1 | 2.1 | 4.0 | 1.1 | | |
| MTT | 46.2±0.8 | 65.4±0.7 | 71.6±0.2 | 24.3±0.3 | 39.7±0.4 | 47.7±0.2 | 8.8±0.3 | 23.2±0.2 |
| +DRUPI | **47.4±0.5** | **65.8±0.6** | **71.7±0.2** | **25.6±0.4** | **40.8±0.3** | 48.8±0.3 | **11.2±0.1** | **24.9±0.2** |
| ↑ | 1.2 | 0.4 | 0.1 | 1.3 | 1.1 | 1.1 | 2.4 | 1.7 |
| Full Dataset | 84.8±0.1 | | | 56.2±0.3 | | | 37.6±0.4 | |

For coreset selection, we benchmarked against several representative baselines, including Random, L-conf, Entropy, Margin (Coleman et al., 2019a), Glister (Killamsetty et al., 2021b), Graig (Mirzasoleiman et al., 2020), Herding (Welling, 2009), k-Center (Har-Peled & Mazumdar, 2004), and Forgetting (Toneva et al., 2018). More detailed settings are provided in Appendix A.2.

For dataset distillation, we evaluated a range of advanced methods, including KIP (Nguyen et al., 2020), DM (Zhao & Bilen, 2022), DSA (Zhao & Bilen, 2021), DCC, DSAC (Lee et al., 2022), CAFE (Wang et al., 2022), IDM (Zhao et al., 2023), DC (Zhao et al., 2020), and MTT (Cazenavette et al., 2022). In line with prior studies, we used networks with instance normalization as the default setting. Unless otherwise specified, distillation was performed with a depth-3 ConvNet for CIFAR-10/100, a depth-4 ConvNet for Tiny ImageNet, and a depth-5 ConvNet for ImageNet subsets. See Appendix A.3 for more details.

It is worth noting that for both pruning and distillation methods, we initialized all data-label pairs using the baseline method and employed a weakly-trained model (trained for one epoch) to extract feature labels, which were then used to synthesize privileged information. By default, we utilized DC to synthesize one feature label per data-label pair, aligning it with features extracted from the final layer of a ConvNet during bi-level optimization. We employed $\lambda_{reg} = 0.5$ and $\lambda_{task} = 0.1$, and learning rate = 0.1 for feature optimization.

## 4.2 MAIN RESULTS

**Coreset selection.** In our experiments, DRUPI utilizes the reduced dataset initialized with the Herding, k-Center, and Forgetting methods to assess its performance across diverse fraction on CIFAR-10/100. As shown in Table 1, by incorporating privileged information, these methods consistently

Table 3: Results on ImageNet subsets when integrating DRUPI into dataset distillation methods. Reduced datasets are initialized with MTT, and feature labels are synthesized with MTT.

| Dataset | ImageNette | | ImageWoof | | ImageFruit | | ImageMeow | | ImageYellow | |
|---|---|---|---|---|---|---|---|---|---|---|
| IPC | 1 | 10 | 1 | 10 | 1 | 10 | 1 | 10 | 1 | 10 |
| MTT | 47.7±0.9 | 63.0±1.3 | 28.6±0.8 | 35.8±1.8 | 26.6±0.8 | 40.3±1.3 | 30.7±1.6 | 40.4±2.2 | 45.2±0.8 | 60.0±1.5 |
| +DRUPI | **50.5±0.1** | **65.7±0.5** | **31.3±0.2** | **37.5±1.0** | **29.1±1.4** | **43.0±0.9** | **34.0±1.6** | **43.8±0.9** | **46.6±0.6** | **62.2±0.9** |
| ↑ | 2.8 | 2.7 | 2.7 | 2.7 | 2.5 | 2.7 | 3.3 | 3.4 | 1.4 | 2.2 |

outperformed the baseline across a range of fraction settings on CIFAR-10/100. Particularly, on the CIFAR-10 (fraction = 0.4%) , DRUPI achieved a performance increase of 24.4% on the Forgetting method and 24.3% on the Herding method. We find that for datasets without optimized instances (*e.g.*, selected coresets), the performance gain is much higher than those with optimized samples.

**Dataset distillation.** We employed DRUPI in several classical dataset distillation methods, where privileged information is obtained with DC and MTT. Table 2 summarizes the classification performances of ConvNets trained with different distillation methods. Specifically, applying DRUPI to DC on CIFAR-100 (10 IPC) resulted in a 4% improvement. For MTT, DRUPI delivered a 2.4% gain on Tiny ImageNet (1 IPC). Additionally, we evaluated its effectiveness on ImageNet subsets, as shown in Table 3, where DRUPI applied to MTT led to a 3.4% improvement on ImageMeow with 10 IPC, demonstrating strong performance even on larger datasets. Notably, learning both feature labels with DRUPI outperforms simply extracting features alone. Further results for DATM are provided in Appendix B.1.

Table 4: Cross-architecture evaluation of coresets on unseen networks. Reduced datasets are initialized with different pruning methods on CIFAR-10 (0.2%). Feature labels are learned with DC. We utilized ConvNet for synthesizing feature labels.

| | LeNet | MLP | ResNet | VGG | ConvNet | AlexNet |
|---|---|---|---|---|---|---|
| Herding | 23.0±1.3 | 21.3±0.4 | 26.2±0.9 | 24.1±0.7 | 25.1±0.5 | 23.3±1.3 |
| +DRUPI | 32.4±1.9 | 30.4±0.4 | 36.9±1.0 | 36.4±0.7 | 46.3±0.6 | 33.2±2.0 |
| ↑ | 9.4 | 9.1 | 10.7 | 12.3 | 21.1 | 9.9 |
| Forgetting | 25.5±1.6 | 23.8±0.3 | 24.7±1.0 | 21.0±0.4 | 24.0±0.5 | 25.5±0.9 |
| +DRUPI | 35.0±1.6 | 33.1±0.5 | 38.1±0.9 | 35.3±0.6 | 47.1±0.7 | 35.7±1.1 |
| ↑ | 9.5 | 9.3 | 13.4 | 14.3 | 23.1 | 10.2 |
| k-Center | 21.6±1.1 | 19.7±0.4 | 24.0±0.8 | 21.5±0.8 | 23.1±0.7 | 21.3±0.7 |
| +DRUPI | 34.4±1.3 | 31.4±0.3 | 36.0±1.5 | 34.3±0.6 | 46.5±0.6 | 36.0±1.6 |
| ↑ | 12.8 | 11.7 | 12.0 | 12.8 | 23.4 | 14.7 |

Table 5: Cross-architecture evaluations of distilled datasets on unseen networks. Reduced datasets are initialized with DC on CIFAR-10 (IPC=10). Feature labels are learned with DC.

| Train\Test | DC | +DRUPI | DC | +DRUPI | DC | +DRUPI |
|---|---|---|---|---|---|---|
| | LeNet | | ConvNet | | ResNet | |
| LeNet | 23.3±5.3 | **28.8±4.1** (5.5) | 35.8±0.6 | **46.6±0.7** (10.8) | 29.7±2.0 | **36.1±1.3** (6.4) |
| MLP | 28.0±1.1 | **28.5±4.1** (0.5) | 29.0±1.1 | **46.5±0.8** (17.6) | 21.8±1.8 | **36.5±1.2** (14.7) |
| ResNet | 22.0±1.5 | **25.9±2.6** (3.9) | 36.2±0.8 | **45.9±0.6** (9.7) | 33.6±1.3 | **36.1±1.8** (2.5) |
| VGG | 22.7±2.4 | **28.3±1.3** (5.6) | 33.5±1.1 | **46.4±0.6** (12.9) | 17.5±1.9 | **35.8±1.5** (18.3) |
| ConvNet | 16.8±1.6 | **24.7±4.3** (7.9) | 44.5±0.9 | **47.1±0.9** (2.6) | 36.2±1.6 | **37.5±1.1** (1.3) |
| AlexNet | 32.2±1.9 | **34.6±1.6** (2.4) | 36.9±0.9 | **46.2±0.6** (9.3) | 31.5±1.5 | **36.1±1.4** (4.6) |
| Train\Test | VGG | | MLP | | AlexNet | |
| LeNet | 34.0±0.7 | **34.9±0.7** (0.9) | 32.6±0.5 | **34.3±0.5** (1.7) | 25.6±2.1 | **32.4±2.9** (6.8) |
| MLP | 22.3±1.0 | **34.6±0.7** (12.3) | 36.6±0.5 | **37.1±0.4** (0.5) | 29.9±2.5 | **30.0±0.9** (0.1) |
| ResNet | 31.6±0.9 | **32.6±0.7** (1.0) | 28.3±0.9 | **30.0±0.6** (1.7) | 24.4±3.3 | **27.2±3.6** (2.8) |
| VGG | 30.1±0.7 | **34.5±0.9** (4.4) | 27.0±0.7 | **31.8±1.1** (4.8) | 22.5±2.5 | **27.1±3.2** (4.6) |
| ConvNet | 35.4±0.6 | **35.8±0.8** (0.4) | 28.0±0.8 | **32.3±1.0** (4.3) | 20.1±4.5 | **27.3±3.2** (7.2) |
| AlexNet | 33.8±0.7 | **34.3±0.8** (0.5) | 31.0±0.6 | **32.3±0.5** (1.3) | 34.0±4.6 | **38.2±1.7** (4.2) |

## 4.3 CROSS-ARCHITECTURE GENERALIZATION

Cross-architecture evaluation is a critical step toward ensuring robust generalization across previously unseen architectures. We measured the quality of reduced dataset with privileged information on both pruning and distillation settings. To address the misalignment between the shapes of the learned feature/attention labels and the intermediate features of different network architectures, we introduced an additional fully connected layer that is trained alongside the evaluation model.

For pruning methods, we utilized ConvNet to synthesize feature labels for selected coresets, and benchmarked their performance across 6 distinct network architectures. As illustrated in Table 4, on the CIFAR-10 (fraction = 0.2%), all 3 methods yielded performance gains exceeding 20% on ConvNet. These methods consistently exhibit improvements exceeding 10% in most cases.

For distillation methods, we applied DRUPI to synthesize feature labels for reduced datasets initialized by DC and MTT. As detailed in Table 5, we conducted experiments on distilled dataset initialized by DC on CIFAR-10 (10 IPC), and trained models on 6 distinct network architectures, and evaluated the model performance across them. Notably, DRUPI achieved an 18.3% performance improvement over the baseline when training on VGG and evaluating on ResNet. Additional results on attention labels are provided in Appendix B.2. Table 9 presents the results of applying DRUPI to MTT on CIFAR-10 (1 IPC), showing significant gains, such as an 11.1% improvement when training on ConvNet and evaluating on AlexNet.

## 5 DISCUSSION

**Different ways for synthesizing privileged information.** As discussed in Section 3.2, there are multiple methods for synthesizing feature labels. A straightforward approach is to assign feature labels using a pre-trained model. However, as shown in Figure 4, this can sometimes degrade dataset quality. In contrast, learning feature labels offers greater flexibility and adaptability. As demonstrated in Figure 4(a), reduced datasets with learned feature labels significantly outperform those with directly assigned features. This is because feature labels extracted from a pre-trained model often lead to overly discriminative features with low diversity. The empirical results also support the observation that overly discriminative feature labels with strong task supervision can hurt performance in Figure 3. More detailed results are provided in Table 10 in Appendix B.3.

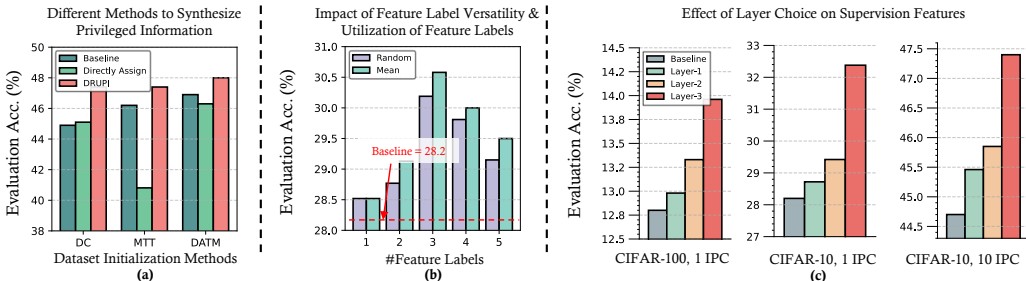

Figure 4: (a) Comparison of different methods for obtaining feature labels in datasets initialized with various distillation methods. Our results indicate that learning-based methods yield the best performance. (b) Impact of feature label versatility and the utilization of multiple feature labels. We find that incorporating more feature labels produces a more robust reduced dataset, with averaging the features outperforming random selection. (c) Evaluation of supervision using different layers from a depth-3 ConvNet for synthesizing feature labels. Results show that, across different IPCs and datasets, using the final layer features for supervision generates the most effective reduced dataset.

**Impact of feature label versatility and methods for utilizing feature labels.** We investigated the effect of synthesizing multiple feature labels for a single data-label pair. As shown in Figure 4(b), experiments demonstrate that increasing the number of feature labels enhances performance, likely due to the greater versatility captured by additional features. However, too many feature labels for a single input can introduce excessive diversity, leading to degraded performance. This verifies the trade-off between the diversity and discriminability of feature labels. Furthermore, averaging multiple feature labels outperforms random selection, which enables us to save only the averaged feature labels. Hence, increasing the number of feature labels does not bring more storage overhead.

**Layer choice for supervision features.** We conducted an in-depth analysis to determine which ConvNet layer's features are most effective for supervision. Specifically, we compared features extracted from the first, second, and final layers of a depth-3 ConvNet. Figure 4(c) shows that deeper layers consistently yielded better performance. This is likely due to the final layer's ability to capture more complex and discriminative information, effectively representing high-level semantics. Therefore, we used the last layer's features to supervise the synthesis of feature labels by default.

For more details on different methods of initializing feature labels, please refer to Appendix B.4. Ablation studies on the regression magnitude of MSE are provided in Appendix C.1. Additionally, we explored the potential of feature regression using losses beyond MSE in Appendix C.2.

## 6 CONCLUSION

In this paper, we introduced DRUPI, a novel framework that synthesizes privileged information for reduced datasets. To the best of our knowledge, DRUPI is the first approach to go beyond the traditional data-label paradigm by utilizing synthesized feature labels. Extensive experiments on ImageNet, CIFAR-10/100, and Tiny ImageNet validate the effectiveness of DRUPI, demonstrating significant improvements in model performance when integrated with existing reduction techniques. Additionally, we showed that achieving a balance between the discriminability and diversity of the synthesized feature labels is crucial for maximizing the quality of the reduced dataset.

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

# A    DETAILED EXPERIMENTAL SETTINGS

## A.1    COMPUTATIONAL RESOURCES

The training was conducted on NVIDIA GPUs, specifically RTX 4090 and A100. All coreset selection experiments were run on A100 GPUs, while all DC-related experiments were carried out with RTX 4090. For MTT and DATM experiments on CIFAR-100 with IPC 10 and 50, as well as Tiny ImageNet, we utilized four NVIDIA A100 GPUs. Results of smaller datasets and lower IPC settings were conducted with RTX 4090.

## A.2    CORESET SELECTION

We first employed several coreset selection methods to initialize our reduced dataset, specifically using Herding (Welling, 2009), k-Center (Har-Peled & Mazumdar, 2004), and Forgetting (Toneva et al., 2018), where each data point was selected based on scores from a pre-trained ResNet-18. Next, we synthesized feature labels for the coreset using DC Zhao et al. (2020), assigning these labels to the intermediate features of a ConvNet trained for just one epoch. We also fine-tuned the initial images using the same method as for feature label learning, although we found that simply synthesizing feature labels could already bring performance improvement for reduced datasets.

Then we provide detailed settings for hyper-parameters. The hyper-parameters include:

- $\lambda_{reg}$: regularization coefficient, which controls the strength of the regularization term in the loss function.
- $\lambda_{task}$: task supervision coefficient, determining the discriminative power of the synthesized feature labels.
- $n_{feat}$: number of feature labels synthesized for a single data-label pair, which controls the diversity.

Unless otherwise specified, for the CIFAR-10 dataset, we set $\lambda_{reg}$ to 0.5, while for CIFAR-100, it was set to 5. Across all configurations, the task supervision coefficient $\lambda_{task}$ was set to 0.001, and we only synthesized one feature label for a single data-label pair ($n_{feat} = 1$) in the reduced dataset.

## A.3    DATASET DISTILLATION

For dataset distillation methods, we initialized the reduced datasets (images and labels) using distilled datasets and applied the same distillation method to synthesize feature labels. Specifically, we used DC (Zhao et al., 2020), MTT (Cazenavette et al., 2022), and DATM (Guo et al., 2023) for both data-label initialization and feature label synthesis.

We followed the original image-label synthesis settings of these distillation methods to generate feature labels. Table 6 provide the detailed hyperparameter settings used for experiments on CIFAR-10, CIFAR-100, Tiny ImageNet. Table 7 provides the parameter settings for the MTT method on the ImageNet subsets. By default, we set $n_{feat}$ to 1 and $\lambda_{reg}$ to 0.01. We explored different configurations of Images Per Class (IPC), specifically IPC = $\{1, 10, 50\}$ for CIFAR-10 and CIFAR-100, and IPC = $\{1, 10\}$ for Tiny ImageNet.

For each method, the hyperparameters for feature synthesis are fine-tuned across different datasets and IPC settings to achieve optimal performance. For instance, in CIFAR-10 (1 IPC), the DC method utilizes $\lambda_{reg} = 1.5$, $\lambda_{task} = 0.1$, and $n_{feat} = 1$. In contrast, for 50 IPC, the same method adjusts its hyperparameters to $\lambda_{reg} = 0.001$, $\lambda_{task} = 0.005$, and $n_{feat} = 1$. Similar fine-tuning is performed for all datasets and IPC values across each method.

For distilled datasets with feature labels, we also tried to incorporate additional forms of privileged information, such as soft labels, to further enrich the privileged information in the synthetic dataset. Specifically, we used a pre-trained network to generate soft labels for the distilled dataset. Some methods like DATM have already learned soft labels. We only synthesized soft labels for DC and MTT based reduced datasets. We provide futher results on soft labels in Table 12.

Table 6: Hyperparameter settings for dataset distillation methods. $^\dagger$ denotes that soft labels were synthesized in this set of experiments to further enrich the privileged information in the synthetic dataset. We used a pre-trained model to synthesize soft labels.

| Dataset | IPC | Method | $\lambda_{reg}$ | $\lambda_{task}$ | $n_{feat}$ |
|---|---|---|---|---|---|
| CIFAR-10 | 1 | DC | 1.5 | 0.1 | 1 |
| | | MTT$^\dagger$ | 0.5 | 0.001 | 1 |
| | | DATM | 0.5 | 0.001 | 5 |
| | 10 | DC | 0.5 | 0.1 | 1 |
| | | MTT | 0.0005 | 0.01 | 3 |
| | | DATM | 0.05 | 0.1 | 3 |
| | 50 | DC | 0.01 | 0.01 | 1 |
| | | MTT | 0.05 | 0.001 | 1 |
| | | DATM | 0.05 | 0.001 | 1 |
| CIFAR-100 | 1 | DC | 1.5 | 0.1 | 1 |
| | | MTT | 0.5 | 0.01 | 3 |
| | | DATM | 0.05 | 0.01 | 1 |
| | 10 | DC$^\dagger$ | 0.001 | 0.005 | 1 |
| | | MTT | 0.005 | 0.001 | 1 |
| | | DATM | 0.05 | 0.001 | 1 |
| | 50 | DC$^\dagger$ | 0.5 | 0.1 | 1 |
| | | MTT$^\dagger$ | 0.5 | 0.01 | 3 |
| | | DATM | 0.0005 | 0.001 | 1 |
| Tiny ImageNet | 1 | MTT$^\dagger$ | 0.5 | 0.0001 | 3 |
| | | DATM | 0.005 | 0.001 | 1 |
| | 10 | MTT$^\dagger$ | 0.005 | 0.001 | 1 |
| | | DATM | 0.005 | 0.001 | 1 |

Table 7: Hyperparameter settings for ImageNet subsets used in MTT experiments.

| Dataset | IPC | $\lambda_{reg}$ |
|---|---|---|
| ImageNette | 1 | 0.005 |
| | 10 | 0.005 |
| ImageWoof | 1 | 0.005 |
| | 10 | 0.5 |
| ImageFruit | 1 | 0.005 |
| | 10 | 0.5 |
| ImageMeow | 1 | 0.00005 |
| | 10 | 0.5 |
| ImageYellow | 1 | 0.005 |
| | 10 | 0.5 |

# B ADDITIONAL RESULTS ON DRUPI

## B.1 FURTHER PERFORMANCE RESULTS

In Section 4.2, we presented results for two dataset distillation methods, DC and MTT. Here, we provide the results for DATM on the CIFAR-10 and CIFAR-100 datasets. We initialized the images and labels with DATM and used it to synthesize feature labels for the reduced dataset. Experiments were conducted using a ConvNet for both distillation and evaluation tasks. We additionally provide baselines such as (Zhou et al., 2022; Loo et al., 2023; Cui et al., 2023; Du et al., 2023; Guo et al., 2023) .Table 8 shows the performance improvements of DRUPI over various existing methods under different IPC settings. Specifically, the datasets used in these experiments include CIFAR-10 with IPC = {1, 10, 50} and CIFAR-100 with IPC = {1, 10}. Each dataset was further evaluated with varying data fractions, allowing us to assess the generalization of the methods across different data availability scenarios.

Table 8: The application of DRUPI to DATM across CIFAR-10, CIFAR-100, accompanied by a comparative analysis with existing methods. ConvNet is utilized for both distillation and evaluation. Our methodology demonstrates enhanced performance compared to previous results.The ↑ symbol signifies performance enhancements compared to random selection.

| Dataset | CIFAR-10 | | | CIFAR-100 | |
|---|---|---|---|---|---|
| IPC | 1 | 10 | 50 | 1 | 10 |
| Fraction (%) | 0.02 | 0.2 | 1 | 0.2 | 2 |
| Random | $15.4_{\pm0.3}$ | $31.0_{\pm0.5}$ | $50.6_{\pm0.3}$ | $4.2_{\pm0.3}$ | $14.6_{\pm0.5}$ |
| KIP | $49.9_{\pm0.2}$ | $62.7_{\pm0.3}$ | $68.6_{\pm0.3}$ | $15.7_{\pm0.2}$ | $28.3_{\pm0.1}$ |
| FRePo | $46.8_{\pm0.7}$ | $65.5_{\pm0.4}$ | $71.7_{\pm0.4}$ | $28.7_{\pm0.2}$ | $42.5_{\pm0.4}$ |
| RCIG | $53.9_{\pm1.0}$ | $69.1_{\pm0.4}$ | $73.5_{\pm0.3}$ | $39.3_{\pm0.4}$ | $44.1_{\pm0.4}$ |
| DM | $26.0_{\pm0.8}$ | $48.9_{\pm0.6}$ | $63.0_{\pm0.4}$ | $11.4_{\pm0.3}$ | $29.7_{\pm0.3}$ |
| DSA | $28.8_{\pm0.7}$ | $52.1_{\pm0.5}$ | $60.6_{\pm0.5}$ | $13.9_{\pm0.3}$ | $32.3_{\pm0.3}$ |
| DCC | $32.9_{\pm0.8}$ | $49.4_{\pm0.5}$ | $61.6_{\pm0.4}$ | $13.3_{\pm0.3}$ | $30.6_{\pm0.4}$ |
| DSAC | $34.0_{\pm0.7}$ | $54.5_{\pm0.5}$ | $64.2_{\pm0.4}$ | $14.6_{\pm0.3}$ | $33.5_{\pm0.3}$ |
| CAFE | $30.3_{\pm1.1}$ | $46.3_{\pm0.6}$ | $55.5_{\pm0.6}$ | $12.9_{\pm0.3}$ | $27.8_{\pm0.3}$ |
| IDM | $45.6_{\pm0.7}$ | $58.6_{\pm0.1}$ | $67.5_{\pm0.1}$ | $20.1_{\pm0.3}$ | $45.1_{\pm0.1}$ |
| TESLA | $\mathbf{48.5_{\pm0.8}}$ | $66.4_{\pm0.8}$ | $72.6_{\pm0.7}$ | $24.8_{\pm0.4}$ | $41.7_{\pm0.3}$ |
| FTD | $46.0_{\pm0.4}$ | $65.3_{\pm0.4}$ | $73.2_{\pm0.2}$ | $24.4_{\pm0.4}$ | $42.5_{\pm0.2}$ |
| DATM | $46.9_{\pm0.5}$ | $66.8_{\pm0.2}$ | $76.1_{\pm0.3}$ | $27.9_{\pm0.2}$ | $47.2_{\pm0.4}$ |
| +DRUPI | $48.0_{\pm0.4}$ | $\mathbf{67.8_{\pm0.3}}$ | $\mathbf{76.4_{\pm0.3}}$ | $\mathbf{28.4_{\pm0.5}}$ | $\mathbf{47.6_{\pm0.2}}$ |
| ↑ | 1.1 | 1.0 | 0.3 | 0.5 | 0.4 |
| Full Dataset | $84.8_{\pm0.1}$ | | | $56.2_{\pm0.3}$ | |

## B.2 RESULTS ON CROSS-ARCHITECTURE EVALUATION WITH ATTENTION LABELS

In addition to the results presented in Table 4 and 5 of the main paper, which summarize the performance of utilizing the synthesized datasets from pruning and dataset distillation to initialize the feature labels, we provide additional results in Table 9 to evaluate the cross-architecture generalization by using both feature labels and attention labels on different reduced datasets.

Table 9 illustrates the cross-architecture evaluations conducted for both dataset condensation (DC) and MTT across various network architectures. In these experiments, we propose DRUPI-F (feature labels learned by DC) and its variant DRUPI-A, which is further enhanced by pooling through spatial attention. Similarly, feature labels synthesized with MTT are also average pooled into channel attention labels. We utilized several networks for cross evaluatoin, including LeNet, MLP, ResNet, VGG, ConvNet, and AlexNet.

We provide examples to illustrate the operation. For example, the first layer feature of a depth-3 ConvNet can be used to supervise the learning of feature labels. In this case, a single feature label takes the form of $Ch \times H \times W$ (*e.g.*, $128 \times 16 \times 16$), which is reduced to a $128 \times 1 \times 1$ channel attention label after average pooling. Attention labels provide a more efficient way for using privileged information.

Additionally, during the cross-architecture process, if the features obtained from the training and testing models differ in shape or dimensionality, we employ an additional fully connected (FC) layer to align the features. For example, if the feature representation from the training model has a shape of $128 \times 16 \times 16$ and the testing model's feature representation is $64 \times 16 \times 16$, the FC layer reshapes the $128 \times 16 \times 16$ feature into the $64 \times 16 \times 16$ format. The FC takes the input feature from the source architecture and transforms it into a format compatible with the target architecture by adjusting the dimensionality of the feature space.

The table demonstrates that initializing the reduced dataset with feature labels assigned through intermediate features of pre-trained networks leads to significant performance improvements across all architectures. Further gains are observed when channel attention pooling is applied. By leveraging Eq. (5) to learn and update the feature labels, the reduced dataset consistently yields competitive or improved performance across different networks. This highlights the benefits of feature and attention labels in enhancing model generalization, with particularly notable improvements observed in ConvNet and AlexNet settings. Channel attention pooling further contributes to these gains, reinforcing the effectiveness of this approach across architectures.

Table 9: Cross-architecture performance comparison using feature labels and channel attention labels for initialization and learning on reduced datasets. We utilized reduced datasets initialized by both the DC and MTT methods. DC was employed for learning both feature labels and attention labels, while MTT was used exclusively for attention label learning through average pooling.

|  | LeNet | MLP | ResNet | VGG | ConvNet | AlexNet |
|---|---|---|---|---|---|---|
| DC | $16.8_{\pm5.2}$ | $28.0_{\pm0.6}$ | $36.2_{\pm1.6}$ | $35.4_{\pm0.6}$ | $44.5_{\pm0.5}$ | $20.1_{\pm4.5}$ |
| +DRUPI-F | $24.7_{\pm4.3}$ | $32.3_{\pm1.0}$ | $37.5_{\pm1.8}$ | $35.8_{\pm0.8}$ | $47.1_{\pm0.9}$ | $27.3_{\pm3.2}$ |
| +DRUPI-A | $23.7_{\pm6.6}$ | $27.9_{\pm0.5}$ | $37.7_{\pm1.1}$ | $35.6_{\pm0.7}$ | $45.6_{\pm0.5}$ | $25.3_{\pm3.0}$ |
| MTT | $29.1_{\pm1.5}$ | $29.1_{\pm0.4}$ | $35.1_{\pm1.1}$ | $31.4_{\pm0.9}$ | $45.6_{\pm0.7}$ | $24.0_{\pm0.8}$ |
| +DRUPI-A | $31.6_{\pm1.7}$ | $29.4_{\pm0.5}$ | $36.7_{\pm5.2}$ | $37.9_{\pm2.8}$ | $47.0_{\pm0.7}$ | $34.1_{\pm4.4}$ |

## B.3 ADDITIONAL RESULTS ON DIFFERENT METHODS FOR SYNTHESIZING PRIVILEGED INFORMATION

As a complement to Figure 4(a), Table 10 provides a more comprehensive overview of the experimental results on different methods for synthesizing privileged information. Specifically, it presents a detailed comparison between direct feature label assignment and the DRUPI framework across three dataset distillation methods: DC, MTT, and DATM. These evaluations were conducted on CIFAR-10, CIFAR-100, and Tiny ImageNet datasets with varying images per class (IPC) values, and include results for different fractions of the full dataset.

For each method, the results are provided for various IPC settings, showing the performance both with direct feature assignment and with the inclusion of DRUPI (feature label). The DRUPI framework consistently improves performance across different methods and datasets, as indicated by the higher accuracy values. Specifically, DRUPI shows substantial improvements over direct assignment, particularly in the low-data regime, such as CIFAR-100 1 IPC and Tiny ImageNet 1 IPC. These findings underscore the effectiveness of DRUPI in enhancing the generalization and performance of dataset distillation techniques, even with limited data.

Table 10: Experimental results comparing direct feature label assignment and the DRUPI framework across three dataset distillation methods (DC, MTT, DATM) on various datasets with different IPC values.

| Dataset | CIFAR-10 | | | CIFAR-100 | | | Tiny ImageNet |
|---|---|---|---|---|---|---|---|
| IPC | 1 | 10 | 50 | 1 | 10 | 50 | 1 |
| Fraction (%) | 0.02 | 0.2 | 1 | 0.2 | 2 | 10 | 0.2 |
| DC | $28.3_{\pm0.5}$ | $44.9_{\pm0.5}$ | $53.9_{\pm0.5}$ | $12.8_{\pm0.3}$ | $25.2_{\pm0.3}$ | $29.8_{\pm0.3}$ | - |
| Directly Assign | $28.3_{\pm0.8}$ | $45.1_{\pm0.5}$ | $54.1_{\pm0.4}$ | $12.7_{\pm0.4}$ | $25.1_{\pm0.4}$ | $29.8_{\pm0.4}$ | - |
| DRUPI | $31.5_{\pm0.9}$ | $47.4_{\pm0.9}$ | $55.0_{\pm0.5}$ | $14.9_{\pm0.4}$ | $29.2_{\pm0.5}$ | $30.9_{\pm0.5}$ | - |
| MTT | $46.2_{\pm0.8}$ | $65.4_{\pm0.7}$ | $71.6_{\pm0.2}$ | $24.3_{\pm0.3}$ | $39.7_{\pm0.4}$ | $47.7_{\pm0.2}$ | $8.8_{\pm0.3}$ |
| Directly Assign | $40.8_{\pm1.8}$ | $56.2_{\pm1.1}$ | $66.1_{\pm0.5}$ | $23.9_{\pm0.5}$ | $38.1_{\pm0.4}$ | $47.4_{\pm0.2}$ | $8.1_{\pm0.4}$ |
| DRUPI | $47.4_{\pm0.5}$ | $65.8_{\pm0.6}$ | $71.7_{\pm0.2}$ | $25.6_{\pm0.4}$ | $40.8_{\pm0.3}$ | $47.9_{\pm0.3}$ | $11.0_{\pm0.1}$ |
| DATM | $46.9_{\pm0.5}$ | $66.8_{\pm0.2}$ | $76.1_{\pm0.3}$ | $27.9_{\pm0.2}$ | $47.2_{\pm0.4}$ | $55.0_{\pm0.2}$ | $17.1_{\pm0.3}$ |
| Directly Assign | $46.3_{\pm0.7}$ | $64.5_{\pm0.5}$ | $73.7_{\pm0.4}$ | $26.6_{\pm0.3}$ | $36.2_{\pm0.5}$ | $55.5_{\pm0.3}$ | $15.6_{\pm0.1}$ |
| DRUPI | $48.0_{\pm0.4}$ | $67.8_{\pm0.3}$ | $76.4_{\pm0.3}$ | $28.4_{\pm0.5}$ | $47.6_{\pm0.2}$ | $55.0_{\pm0.1}$ | $17.6_{\pm0.1}$ |
| Full Dataset | | $84.8_{\pm0.1}$ | | | $56.2_{\pm0.3}$ | | $37.6_{\pm0.4}$ |

## B.4 EFFECTS ON FEATURE LABEL INITIALIZATION

We further explored the issue of feature label initialization. Specifically, initialization can be performed either by using random noise or by feeding synthetic images into a ConvNet to extract intermediate layer features for initialization.

Figure 5 presents a performance comparison between two feature initialization approaches—random noise and assigned features—across the CIFAR-10 and CIFAR-100 datasets with varying images per class (IPC) settings. In this comparison, noise (yellow bars) refers to randomly initialized feature

labels, while real (blue bars) refers to initialization using features extracted from synthetic datasets passed through the network.

For CIFAR-10 1 IPC setting, the initialization with assigned features method achieves significantly higher performance (around 33%) compared to the noise-based initialization (approximately 30%). A similar trend is observed in the CIFAR-10 (10 IPC) case, where initialization with assigned features substantially outperforms noise.

On the CIFAR-100 (1 IPC) dataset, initialization with assigned features also demonstrates better performance, with a clear margin over noise initialization, where using assigned features reaches around 29% while noise remains below 28%. The same pattern holds for CIFAR-100 (10 IPC), where initialization with assigned features achieves a accuracy of over 47%, far exceeding the noise initialization's performance of approximately 45%.

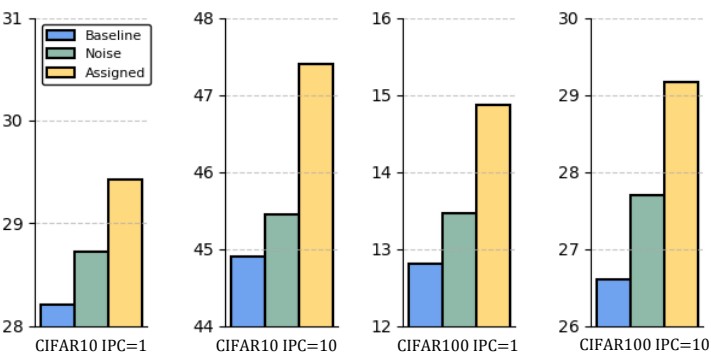

Figure 5: Comparison of noise initialization (yellow) and initialization with assigned features (blue) from a pre-trained ConvNet on CIFAR-10 and CIFAR-100 across different IPC settings.

## C  ABLATION ON REGRESSION SUPERVISION

### C.1  SENSITIVITY OF REGRESSION MAGNITUDE

We further investigated the influence of $\lambda_{reg}$ on the model's performance. The results in Table 11 demonstrate that varying the MSE regularization parameter ($\lambda_{reg}$) does not significantly impact the final accuracy. While there is a slight improvement as $\lambda_{reg}$ increases, from 28.91% at $\lambda_{reg} = 0.05$ to 30.74% at $\lambda_{reg} = 10$, the overall effect remains relatively small. This suggests that the model's performance is not highly sensitive to the regularization weight for MSE loss within the tested range, indicating that other factors may have a more dominant influence on accuracy. In this case, regularization helps prevent overfitting but does not drastically change the model's ability to generalize within the 1 IPC setting for CIFAR-10.

Table 11: Results on CIFAR-10 1 IPC for different $\lambda_{reg}$.

| $\lambda_{reg}$ | 0.05 | 0.1 | 0.5 | 1 | 5 | 10 |
|---|---|---|---|---|---|---|
| Acc (%) | 28.91 | 29.2 | 30.42 | 30.53 | 30.67 | 30.74 |

### C.2  FURTHER REGRESSION SUPERVISION OBJECTIVES

We extended the use of feature labels by integrating additional supervision mechanisms, including CE loss and InfoNCE loss, to enhance the distillation process. Additionally, we performed ablation studies to evaluate the impact of using soft labels. These components contribute to a more structured and informative representation learning framework.

First, we introduced CE loss and InfoNCE loss to provide direct regularization between the feature labels and the supervision features. In line with previous work, we also incorporated soft labels as privileged information.

The combination of CE loss, InfoNCE loss, and soft labels consistently yielded the best performance across our ablation studies, as shown in Tables 12. We found that incorporating InfoNCE or CE loss marginally improved the quality of the reduced dataset, while soft labels, when combined with feature labels, provided a more significant boost in performance.

Table 12: Ablation study of different components in DRUPI across CIFAR-10 and CIFAR-100 (1 IPC). The table compares the effects of feature labels, CE loss ($\mathcal{L}_{CE}$), contrastive loss ($\mathcal{L}_{Info}$), and soft labels on model performance.

| Feature label | $\mathcal{L}_{CE}$ | $\mathcal{L}_{Info}$ | Soft label | Acc (%) |
|---|---|---|---|---|
| | | | | 28.30 |
| ✓ | | | | 31.13 |
| ✓ | ✓ | | | 31.54 |
| ✓ | | ✓ | | 31.47 |
| ✓ | ✓ | ✓ | | 31.90 |
| ✓ | | | ✓ | 32.28 |

(a) CIFAR-10, 1 IPC

| Feature label | $\mathcal{L}_{CE}$ | $\mathcal{L}_{Info}$ | Soft label | Acc (%) |
|---|---|---|---|---|
| | | | | 12.80 |
| ✓ | | | | 13.96 |
| ✓ | ✓ | | | 14.10 |
| ✓ | | ✓ | | 14.38 |
| ✓ | ✓ | ✓ | | 14.64 |
| ✓ | | | ✓ | 14.86 |

(b) CIFAR-100, 1 IPC

## D    PSEUDO CODE OF DRUPI

Algorithm 1 outlines the process where we initialize the reduced dataset with assigned feature labels based on the synthetic dataset from dataset condensation (DC). Subsequently, we update the reduced dataset by learning the feature labels, which are progressively refined during the training process.

---

**Algorithm 1** Dataset Reduction Using Privileged Information (DRUPI) For DC

---

**Require:** Outer-loop steps $K$, inner-loop steps $T$, synthesized privileged information $\mathcal{PI} = \{f_i^\star\}_{i=1}^m$. Let $\psi(\cdot)$ denote the intermediate output of model $g$, and $g = \psi \circ \kappa$, where $\kappa(\cdot)$ is the classifier component of $g$. And $\sigma(\cdot)$ represents the softmax function.

1: Initialize reduced dataset $\mathcal{D}_{syn} = \{(\tilde{x}_i, \tilde{y}_i)\}_{i=1}^m$ and privileged information $\mathcal{PI}$. The initial feature labels are derived from a pre-trained network applied to the synthetic dataset.

2: **for** $k = 0, \ldots, K - 1$ **do**

3:     Initialize neural network weights $\theta_0$

4:     **for** $t = 0, \ldots, T - 1$ **do**

5:         **for** $c = 0, \ldots, C - 1$ **do**

6:             Sample mini-batches $B_T^c$ and $B_S^c$ from $\mathcal{D}_T$ and $\mathcal{D}_{syn}$

7:             Compute Cross-Entropy loss: $\mathcal{L}_{cls} = \mathbb{E}_{(\tilde{x}_i, \tilde{y}_i) \in B_S^c} [\ell_{ce}(\tilde{y}_i, \sigma(g(\tilde{x}_i; \theta_t)))]$.

8:             Compute Mean Squared Error loss: $\mathcal{L}_{reg} = \mathbb{E}_{(\tilde{x}_i, f_i^\star) \in B_S^c} [\ell_{mse}(f_i^\star, \psi(\tilde{x}_i; \theta_t))]$.

9:             Compute task-oriented loss using Eq. (4)

10:             Update synthetic samples $S_c$ using Eq. (5)

11:         **end for**

12:     **end for**

13: **end for**

14: **Output:** An Optimized and extended dataset represented as $D_{syn}^\star = (\tilde{x}_i, \tilde{y}_i, f_i^\star)$

---

