# OpenReview forum: "DRUPI: Dataset Reduction Using Privileged Information"
_ICLR.cc/2025/Conference — ICLR 2025 Conference Withdrawn Submission_

### Official Review · Reviewer_SLLc · 2024-10-15

**Soundness:** 2
**Presentation:** 3
**Contribution:** 2
**Rating:** 3
**Confidence:** 4

**Summary:**

This paper extends dataset condensation—referred to by the author as dataset reduction, a more general term I prefer, as the author's definition in the introduction does not encompass all dataset condensation methods—from conventional image-label pairs to image-label-privileged information pairs. It provides an in-depth analysis of the effectiveness of this approach in various settings, including different label types (feature and attention labels) and task-oriented synthesis to balance discriminability and diversity. The experiments demonstrate that the algorithm can be integrated into several existing dataset condensation methods.

This approach offers an intriguing extension of feature-based knowledge distillation. As a scholar in the fields of knowledge distillation and dataset condensation, I have naturally considered this idea and attempted to implement it. However, I encountered several challenges that ultimately led me to abandon it. Therefore, I believe the authors must address key challenges related to the introduction of privileged information, particularly regarding generalizability. For instance, does the use of privileged information violate the core principles of dataset condensation? (A similar question arose during the ICLR submission process for paper [1].) Additionally, is the extra storage overhead introduced by privileged information manageable?

All in all, this is an interesting approach and I read the article very carefully. However, this article is missing a lot in terms of experimentation, which I'll point out in a follow-up WEakness.

[1] Scaling Up Dataset Distillation to ImageNet-1K with Constant Memory, ICML, 2022

**Strengths:**

1. The core contribution of this paper is the introduction of additional supervised information, specifically privileged information, alongside input data and soft labels. This paradigm effectively extends the capabilities of the dataset condensation algorithm.

2. The authors validate DRUPI on multiple datasets and demonstrate its integration with several classical dataset condensation algorithms, confirming its generalizability to some extent.

**Weaknesses:**

Overall, the biggest drawback of this work is what I mentioned in Summary, the generalization of this paradigm, does it violate the concept of dataset condensation? Is the additional storage overhead introduced acceptable? Besides that, some expressions in the paper need to be improved.

**Presentation:**

1. The concept of dataset distillation is misunderstood. The authors classify dataset reduction into coreset selection and dataset distillation (line 125). However, this raises the question: how should algorithms that synthesize data without training, such as RDED [1], be categorized? The authors argue that all dataset distillation methods adhere to the distillation paradigm and synthesize only unseen data from the training set. In contrast, coreset selection is restricted to algorithms that select a subset of the original dataset.

2. The authors later make a definitional mistake by assuming that all dataset condensation that requires training is bi-level optimization paradigm, which is false. None of the researches [1,2,3,4,5,6] require bi-level optimization.

3. On line 253, Figure 3(a)(c) is a typo.

4. The theoretical analysis provided by the authors was unclear to me, and I am interested in seeing how other reviewers perceive it. Overall, it did not provide any significant insights and felt as though it was included primarily to meet the expectation of a theoretical component.

[1] On the Diversity and Realism of Distilled Dataset: An Efficient Dataset Distillation Paradigm, CVPR, 2024

[2] Squeeze, Recover and Relabel: Dataset Condensation at ImageNet Scale From A New Perspective, NeurIPS, 2023

[3] Generalized Large-Scale Data Condensation via Various Backbone and Statistical Matching, CVPR, 2024

[4] Information Compensation: A Fix for Any-scale Dataset Distillation, CVPR Workshop, 2024

[5] Elucidating the Design Space of Dataset Condensation, NeurIPS, 2024

[6] Diversity-Driven Synthesis: Enhancing Dataset Distillation through Directed Weight Adjustment, NeurIPS, 2024


**Method**

1. The full paper, including the appendices, does not address the additional storage overhead introduced by the use of privileged information. This omission is significant, as the storage burden imposed by the model's intermediate layer outputs is much greater than that of soft labels. For example, on the full ImageNet-1k dataset, the soft labels for IPC 50 alone amount to tens of gigabytes.

2. The generalizability of DRUPI is a significant concern. The results presented in Table 4 are based solely on experiments using convolutional model. I believe this paradigm may not be applicable when using a Vision Transformer as the teacher model and convolutional model as the student model. Furthermore, I think the paper should detail how the authors solved the gap in the feature map due to dimensional inconsistency.

3. The authors likewise need to clarify whether this paradigm violates the classical paradigm of dataset condensation.

**Experiment**

1. The authors tested DRUPI with only a few algorithms, specifically MTT and DC, which is insufficient. To date, dataset condensation algorithms that require training have consistently outperformed dataset selection algorithms in terms of performance. The authors should consider evaluating DRUPI with additional algorithms, such as gradient matching and distribution matching.

2. In addition, decoupled distillation (as referenced in [1-6]) represents the current state-of-the-art (SOTA) algorithm for large-scale datasets, including the full 224x224 ImageNet-1k and the full 224x224 ImageNet-21k. The authors should build on these algorithms to apply DRUPI. Restricting the experiments to small-scale datasets, such as the subset of ImageNet-1k used in the paper, does not significantly advance the field of dataset condensation. This limitation also hinders the potential for dataset condensation to play a meaningful role in the era of foundation models.

**Questions:**

No

---

### Official Review · Reviewer_y24f · 2024-10-24

**Soundness:** 3
**Presentation:** 3
**Contribution:** 2
**Rating:** 5
**Confidence:** 4

**Summary:**

This paper introduces DRUPI, a novel approach for dataset reduction that synthesizes privileged information, such as feature labels, alongside traditional data-label pairs. By leveraging privileged information, DRUPI aims to improve the efficacy of reduced datasets across various machine-learning tasks. The paper thoroughly evaluates the method on multiple benchmarks, including CIFAR-10/100, Tiny ImageNet, and ImageNet subsets, demonstrating significant performance improvements compared to baseline dataset reduction techniques.

**Strengths:**

The introduction of feature labels as privileged information goes beyond the traditional data-label paradigm, providing additional supervision that improves model generalization and robustness.

The paper conducts an extensive set of experiments across a variety of datasets (CIFAR-10/100, Tiny ImageNet, and ImageNet subsets) and methods (coreset selection and dataset distillation), clearly demonstrating the efficacy of DRUPI in improving performance.

The authors provide a rigorous theoretical framework based on VC theory, which justifies the efficacy of the proposed approach, enhancing the contribution and ensuring the generalizability of the paper.

**Weaknesses:**

DRUPI uses the output of a pre-trained model as privileged information and further applies an MSE loss to enforce consistency between the model’s predictions and the privileged information. This approach shares similarities with knowledge distillation techniques. While the method is straightforward and yields competitive results, it inherently limits the upper bound of DRUPI's performance to the quality of the pre-trained model. Consequently, if the pre-trained model is suboptimal, the privileged information may not fully capture the necessary features, potentially constraining the overall effectiveness of DRUPI.


Given the conceptual overlap between DRUPI and knowledge distillation, it would have been beneficial for the paper to include a direct experimental comparison with established knowledge distillation techniques. Such comparisons would provide more insight into how DRUPI differentiates itself from or improves upon these methods, as both share the goal of leveraging additional information to enhance model performance.

The core innovation of DRUPI lies in the introduction of the output from a pre-trained model and the use of consistency constraints to guide model training. However, this concept has been extensively explored in the domain of knowledge distillation. While DRUPI's application of this idea within the context of dataset reduction is novel, the paper does not sufficiently acknowledge the wealth of prior work in knowledge distillation, where similar strategies have been widely adopted.

**Questions:**

Please see weakness.

---

### Official Review · Reviewer_1pBq · 2024-10-28

**Soundness:** 3
**Presentation:** 3
**Contribution:** 2
**Rating:** 5
**Confidence:** 4

**Summary:**

This paper proposes learnable feature labels in the dataset pruning/distillation task. The feature labels are initialized by intermediate outputs of a pre-trained model and are optimized jointly with synthetic data. On several benchmarks, the proposed method improves baseline methods, especially under the pruning setting.

**Strengths:**

1. All the charts in the text are clear and easy to understand.
2. The theoretical analysis further supports the proposed method
3. The distilled dataset can generalize well across various architectures

**Weaknesses:**

1. Some details of the method are not clarified, especially on the pruning part. The authors described the algorithm in the context of DC but have yet to explain how pruning can incorporate learnable feature labels.
2. Analysis in section 5 is trivial and **doesn't** provide enough insights. More discussion can be included.
3. The scalability on **large** IPCs may not be good enough according to the reported results in Table 2 and Appendix.

For the weaknesses mentioned above, I'll raise the corresponding questions below.

**Questions:**

1. The description of the main method is elaborated based on bi-level optimization distillation methods, DC. However, I'd like to know details of how privileged information is learned with coreset selection methods, as improvements shown in Table 1 are significant. The code in the supplementary material also doesn't include this part. (related to weakness 1)

2. As the synthetic dataset now receives more information when being updated, how does the newly introduced feature label affect the distilled dataset? For example, Figure 3(a) shows the t-SNE visualization of feature labels under different $\lambda_{task}$, and the conclusion is that feature labels should balance diversity and discriminative power. However, how does this affect the data? Some visualization could be made to further discuss the impact of different feature labels on the synthetic data. (related to weakness 2)

3. Since the ultimate objective of dataset reduction is to find a smaller set of data on which the model can achieve similar or even better performance,  and lossless performances have only been seen on large IPCs (DATM[1]), it is recommended to show DRUPI can consistently improve baselines even when IPC goes larger (based on reported results, the improvement brought by DRUPI becomes smaller when IPC increases). (related to weakness 3)

4. This question is for discussion. Authors argue that the data-label format should be deprecated for reduced datasets. To improve this, feature labels are introduced to improve performance, and evaluation models are required to adopt a new training objective.
However, at the end of the day, dataset reduction should focus primarily on the **dataset** itself instead of improving the test accuracy by introducing a lot more information. Thus,  to fill this gap, I am curious about the evaluation performance of the model trained on $D^{*}\_{S}$ following the same evaluation method of MTT/DC (without feature label matching) to see if feature labels improve the distilled dataset's quality.


[1] Towards Lossless Dataset Distillation via Difficulty-Aligned Trajectory Matching, ICLR 2024

---

### Official Review · Reviewer_vyX5 · 2024-10-29

**Soundness:** 2
**Presentation:** 3
**Contribution:** 2
**Rating:** 5
**Confidence:** 4

**Summary:**

This paper introduces a novel method for enhancing dataset reduction by incorporating privileged information, such as feature and attention labels, to improve model training. This approach goes beyond traditional data-label compression, providing additional supervision that leads to better generalization and performance. The method demonstrates significant improvements in datasets like CIFAR-10/100 and ImageNet, outperforming baseline reduction techniques. While the results are promising, the paper does not fully address distillation costs or generalization to tasks like object detection. Overall, DRUPI offers a fresh perspective on dataset reduction with meaningful contributions to the field.

**Strengths:**

[S1] The paper presents the use of privileged information, such as feature and attention labels, to improve dataset reduction, demonstrating performance gains on datasets like CIFAR-10/100 and ImageNet.

[S2] The approach integrates well with existing dataset reduction methods, highlighting its flexibility.

[S3] It is supported by a solid theoretical foundation based on VC theory, which explains how privileged information enhances generalization and learning.

[S4] The paper is well-written and easy to understand.

**Weaknesses:**

[W1] The paper lacks a thorough analysis of computational costs, which is critical for assessing the method’s efficiency. I suggest reporting specific metrics such as training time, memory usage, and storage overhead when generating and storing privileged information. It would be helpful to compare these costs with baseline methods to provide a clear understanding of DRUPI’s performance in terms of resource consumption. This analysis will clarify the computational trade-offs involved in using privileged information versus traditional dataset distillation methods.

[W2] Concerns about generalizability remain, particularly regarding the method's applicability to high-resolution datasets and different model architectures. I recommend testing DRUPI on 256x256 ImageNet-1k and on practical datasets such as clinical or aerial images to address this. Additionally, evaluating its performance on ViT models for cross-architecture generalization would provide further insights into the method's broader applicability across deep architectures.

[W3] While the practical applications of dataset distillation are not explored in depth, I understand this may be beyond the scope of a single paper. However, it would strengthen the paper to include a discussion on DRUPI’s potential applications, particularly in areas like neural architecture search (NAS) or continual learning. If feasible, adding an experiment or case study in one of these domains could demonstrate the method’s utility in real-world scenarios.

[W4] The paper does not include comparisons with key state-of-the-art methods like DREAM, DataDAM, and SeqMatch in Table 2. Including these methods would provide a more comprehensive evaluation of DRUPI’s effectiveness. If there are reasons why these methods were not compared, it would be helpful for the authors to explain this and clarify how DRUPI relates to these recent approaches.

**References**:
- [a] Liu, Yanqing, et al. "DREAM: Efficient dataset distillation by representative matching." Proceedings of the IEEE/CVF International Conference on Computer Vision. 2023.
- [b] Sajedi, Ahmad, et al. "DataDAM: Efficient dataset distillation with attention matching." Proceedings of the IEEE/CVF International Conference on Computer Vision. 2023.
- [c] Yin, Zeyuan, et al. "Squeeze, recover and relabel: Dataset condensation at ImageNet scale from a new perspective." Advances in Neural Information Processing Systems 36 (2024).

**Questions:**

1. Does the method show cross-architecture generalization on ViT models?
2. Can HeLlO support strong performance at higher IPCs, such as 100 or 200?
3. How does the proposed framework perform on downstream tasks like object detection or segmentation?
4. It would be interesting to see an ablation study examining the effects of soft labels, ZCA, and EMA on DRUPI.
5. Could the authors include visualizations of some synthetic images for different IPCs? I am curious to see how the synthetic images look.

---

### Official Review · Reviewer_oXjw · 2024-11-02

**Soundness:** 2
**Presentation:** 2
**Contribution:** 2
**Rating:** 3
**Confidence:** 5

**Summary:**

This paper explores the potential to synthesize additional information beyond the standard data-label pairs as a learning target to enhance model training. The authors introduce Dataset Reduction Using Privileged Information (DRUPI), which augments dataset reduction by generating privileged information alongside a reduced dataset. This privileged information may include feature labels or attention labels, providing auxiliary supervision to strengthen model learning. The study finds that effective feature labels strike a balance between being too discriminative and too diverse, with a moderate level yielding good results in enhancing the efficacy of the reduced dataset. Extensive experiments on ImageNet, CIFAR-10/100, and Tiny ImageNet demonstrate that DRUPI integrates smoothly with existing dataset reduction techniques, yielding substantial performance improvements.

**Strengths:**

1.	The use of feature labels and attention labels as privileged information is straightforward and easy to implement.

**Weaknesses:**

1.	The paper does not sufficiently acknowledge prior work, for instance, it should compare and discuss the proposed method with previous dataset distillation methods that use data-soft label structures, which are similar in approach.
2.	The novelty of the privileged information (feature and attention labels derived from feature labels) is limited and similar to approaches such as Re-labeling ImageNet and FKD.
3.	The theoretical analysis using VC theory is general and not specific to the proposed method, making it insufficient to substantiate the approach.
4.	The baselines in Table 1 are weak. In Table 2 of many cases, only marginal improvements are shown over prior methods. Additionally, the lack of results on full ImageNet weakens the validity of the experiments and the effectiveness.
5.	Baseline approaches like DC and MTT are somewhat outdated, newer stronger methods such as SRe2L and RDED are not included in the comparison.

**Questions:**

1.	Since the attention label is average pooled from the feature label, are these two labels redundant?
2.	Equation 6 lacks detail on how feature labels are used in training. The statement, "we apply the same pooling strategy for intermediate features of the given model to calculate the MSE loss between the intermediate features and given feature labels." is unclear and seems to be missing context.
3.	The theoretical analysis relies on existing VC theory without specifics related to the proposed method.

---

### Note · Authors · 2024-11-13

**Comment:**

Dear ACs and Reviewers,

I would like to express my deepest gratitude to the reviewers and editorial team for their valuable feedback and constructive suggestions. Their comments have provided me with insightful guidance and have highlighted areas where the manuscript can be further improved. I truly appreciate the time and effort invested in reviewing my work.

After careful consideration, I have decided to withdraw the manuscript to allow for more revisions that will strengthen its quality and impact. This will enable me to address the reviewers’ comments thoroughly and to incorporate additional improvements for future submissions.

Thank you for your understanding and for providing a platform for scholarly growth and development.

**Withdrawal Confirmation:**

I have read and agree with the venue's withdrawal policy on behalf of myself and my co-authors.